# Comparative evaluation of plasma biomarkers of *Schistosoma haematobium* infection in endemic populations from Burkina Faso

**Mireille Ouedraogo[1,2,3], Jana Christina Hey[4,9], Stan Hilt[5], Veronica Rodriguez Fernandez[1], Doris Winter[4], Ravo Razafindrakoto[6], Pytsje T. Hoekstra[5], Youssouf Kabore[3], Marco Fornili[7], Laura Baglietto[7], Issa Nebie[3], Govert J. van Dam[5], Paul L. A. M. Corstjens[8], Daniela Fusco[4,9], David Modiano[2], Fabrizio Bruschi[1], Valentina D. Mangano[1] ***

1 Department of Translational Research in Medicine and Surgery, University of Pisa, Pisa, Italy,
2 Department of Public Health and Infectious Diseases, University of Rome La Sapienza, Rome, Italy,
3 Centre National de Recherche et Formation sur le Paludisme, Ouagadougou, Burkina Faso, 4 Department of Infectious Diseases Epidemiology, Bernard Nocht Institute for Tropical Medicine, Hamburg, Germany,
5 Leiden University Center for Infectious Diseases, Leiden University Medical Center, Leiden, the Netherlands, 6 Centre d'infectiologie Charles Merieux, Antananarivo, Madagascar, 7 Department of Clinical and Experimental Medicine, University of Pisa, Pisa, Italy, 8 Department of Cell & Chemical Biology, Leiden University Medical Center, Leiden, the Netherlands, 9 German Center for Infection Research, Hamburg—Lübeck—Borstel–Riems, Germany

* valentina.mangano@unipi.it

**Data Availability Statement:** Data can be accessed in the Figshare repository with DOI: 10.6084/m9.figshare.25311364.

## Abstract

Infection with *Schistosoma haematobium* causes urogenital disease associated with organ disfunction, bleeding, pain, and higher susceptibility to infections and cancer. Timely and accurate diagnosis is crucial for prompt and appropriate treatment as well as surveillance efforts, and the use of plasma biomarkers offers important advantages over parasitological examination of urine, including increased sensitivity and the possibility to use the same specimen for multiple investigations. The present study aims to evaluate the diagnostic performance of different plasma biomarkers in endemic populations from Burkina Faso, West Africa. *Schistosoma* spp. Circulating Anodic Antigen (CAA), cell free *S. haematobium* DNA (cfDNA), class M and G antibodies against *S. haematobium* Soluble Worm Antigen Preparation (SWAP) and Soluble Egg Antigen (SEA) were measured in 406 plasma samples. Results of each biomarker test were compared to those of CAA, a Composite Reference Standard (CRS) and Latent Class Analysis (LCA). An identical proportion of positive samples (29%) was observed as a result of CAA and cfDNA testing, with a substantial agreement (84%, Cohen k = 0.62) between the results of the two tests, and a comparable agreement with the results of CRS and LCA. A higher positivity was observed, as expected, as a result of specific antibody testing (47%-72%), with IgG showing a higher agreement than IgM with the three references. Also, higher IgG levels were observed in current vs past infection, and ROC analysis identified optimal cutoff values for improved testing accuracy. This study provides compelling evidence that can inform the choice of the most appropriate diagnostic plasma biomarker for urogenital schistosomiasis in endemic areas, depending

**Funding:** MO was supported by the PhD School in Microbiology, Infectious Diseases and Public Health, Sapienza University of Rome. The funders had no role in study design, data collection and analysis, decision to publish, or preparation of the manuscript.

**Competing interests:** The authors have declared that no competing interests exist.

on the purpose, context, and available resources for testing. Either CAA or cfDNA testing can be used for the diagnosis of patients and for epidemiological investigations, even in absence of urine filtration microscopy, whereas anti-SWAP or anti-SEA IgG can be employed for surveillance and integrated monitoring of control interventions against poverty-associated diseases.

## Author summary

Urogenital schistosomiasis is a chronic debilitating disease affecting populations living in Africa and the Middle East and showing a strong association with poverty. Accurate detection of infection is important both for disease treatment and surveillance. Several tests based on detection in plasma of parasite protein (CAA), parasite DNA or parasite-specific host antibodies (IgM and IgG against SWAP and SEA antigens) are available, and this study aims at comparing them to evaluate their accuracy. The comparison showed that tests based on parasite CAA or DNA yield very similar results and therefore the test of choice for diagnosis or epidemiological investigations can be based on laboratory resources. Additionally, the comparison showed that IgG against SWAP and SEA outperform IgM, and that high accuracy can be achieved by identifying an optimal level to determine positivity (cutoff), making these antibody tests ideal for surveillance purposes.

## Introduction

Human schistosomiasis is a poverty-associated parasitic diseases caused by trematode blood flukes of the genus *Schistosoma*. It is estimated that 250 million people are infected in 78 endemic countries, and that 90% of cases occur in Sub-Saharan Africa (SSA) [1–3].

 *S. haematobium*, endemic in Africa and the Middle East, and since recent years in some areas of Europe [4–6], is the only species causing urogenital disease. Infection occurs by contact with fresh-water bodies infested with free-swimming cercariae larvae released by specific intermediate snail hosts of the genus *Bulinus*. Cercariae penetrate the skin of the human host, shed their fork and become schistosomula, which then migrate via venous circulation to the lungs, the heart and the liver where they mature to the adult stage. Adult worms exit the liver via the portal vein system and reach the venules of the bladder, where they reproduce and gravid females lay eggs. While most eggs remain trapped in venules and body tissues, some are eliminated with urine. When the human host discharges urine in fresh-water bodies, the eggs hatch and release miracidia, which penetrate the snail host and transform into sporocysts. After two generations of sporocysts larval reproduction, these transform into cercariae, able to initiate a new transmission cycle [7–9].

 The disease is mostly accounted by immunopathogenic responses to the worms' eggs accumulated around bladder and ureters resulting in granuloma formation and tissue fibrosis. Clinical manifestations include haematuria, obstructive uropathy, kidney damage and squamous cell carcinoma [7–9]. In females, genital schistosomiasis (FGS) causes ulcerative lesions and fibrosis in vagina, cervix and uterus, resulting in bleeding, pain and higher risk of HIV infection [10, 11]. Control of schistosomiasis mainly focuses on reducing disease burden through periodic, large-scale population treatment with praziquantel, but more comprehensive approaches include water, hygiene and sanitation strategies, health education and snail control [2, 3].

Diagnosis of schistosomiasis is fundamental for appropriate treatment of symptomatic individuals as well as for monitoring the success of control efforts, and a variety of direct (detecting the parasite or parasite molecules) and indirect (detecting the human immune responses to the parasite) laboratory methods are available, each with different advantages and disadvantages. Microscopic examination of urine samples (urine filtration microscopy) for detection of *S. haematobium* eggs has optimal specificity based on distinct morphological features. However, sensitivity is hampered by the irregular timing of eggs release in urine and by their limited number, particularly in low intensity infections, and the examination of at least three specimens is required [12, 13]. Other direct methods include the detection of intestinal antigens regurgitated by *Schistosoma* adult worms and therefore released in circulation, namely Circulating Cathodic Antigen (CCA) and Circulating Anodic Antigen (CAA), in either urine or serum/plasma samples by immunological assays such as ELISA and ICT, including the point-of-care CCA (POC-CCA) test and the upconverting particle lateral flow CAA (UCP-LF CAA) test [14]. Compared to CCA, CAA testing has shown a better sensitivity for the diagnosis of infection by different *Schistosoma* species, including *S. haematobium*, and in areas of low endemicity [15]. It has also been reported that CAA concentration shows little day to day fluctuations, positively correlates with the number of adult worms, and decreases sharply after drug treatment, establishing it as a valuable test for both diagnosis and follow-up [16]. Nonetheless, it must be noted that infection with larval stages cannot be diagnosed by such antigen tests. Detection of cell-free DNA (cfDNA) of schistosomes in either stool/urine or plasma by molecular assays represents a promising method for early direct diagnosis, since it can detect all stages of infection including larval stages, and since optimal analytical sensitivity and specificity can be achieved by choice of multi-copy specific targets [17]. For *S. haematobium*, Real Time PCR protocols based on the detection of *Dra1*, a repeat sequence specific of this species and representing >15% of its genome, have been recently developed and tested in migrant and travellers as well as residents in endemic areas [18–22]. However, cfDNA may circulate for some time after successful treatment and its detection might not therefore always indicate current infection [17]. Regarding indirect methods, these are based on the detection of specific antibodies against parasite antigens in serum/plasma samples by immunological methods such as ELISA, ICT and more recently Luminex [23, 24]. The main advantage of these methods is increased sensitivity compared to urine filtration microscopy, as demonstrated in non-endemic contexts [25, 26]. However, as production of antibodies by plasma cells only starts about 1 week after exposure to a given antigen, a temporal delay is to be expected between infection and the detection of specific antibodies. More importantly, antibodies can still be detectable in plasma after drug treatment and parasite death, both because of antigen persistence and continued exposure to the immune system, and because of the long half-life of class G immunoglobulins (Ig). Therefore, seropositivity might reflect either history of infection or current infection or both. Antigens employed for *S. haematobium* serology have long included soluble excretory/secretory products from the different stages, such as Cercarial Antigen Preparation (CAP), Soluble Worm Adult Preparation (SEA) and Soluble Egg Antigen (SEA) [21,27], and, more recently, recombinant proteins of the tetraspanin family that were identified as promising candidates through immunomic analyses [28, 29]. It is noteworthy that no commercial in vitro diagnostic tests for the specific diagnosis of *S. haematobium* exist to date.

As described above, several assays are available for laboratory diagnosis of *S. haematobium* using plasma samples. The use of plasma samples offers the important advantage of allowing the simultaneous detection of biomarkers of infection with different pathogens in the same specimen. This approach enables increased efficiency not only in the diagnosis of concomitant infections or the differential diagnosis of infections with overlapping clinical features in patients, but also in conducting epidemiological studies and surveillance activities of different

infectious diseases such as malaria and NTDs, including the integrated monitoring of control interventions impact [30–32]. A further advantage is the possibility of retrospective analysis of archived plasma samples for further investigations.

Despite the benefits offered by plasma assays for *S. haematobium* diagnosis, to the best of our knowledge, a comparative evaluation of their diagnostic performance has not yet been conducted in endemic populations. To this aim, we took advantage of archived plasma samples collected during malariological surveys in an area of *S. haematobium* and malaria co-endemicity in Burkina Faso [33].

## Methods

### Ethics statement

Study subjects or their guardians gave written informed consent for participation. The original malaria epidemiological study received approval from the ethical committee of the Ministry of Health of Burkina Faso (2007–048). The use of archived plasma samples for schistosomiasis research received further approval from the ethical committee of the Ministry of Health and Public Hygiene of Burkina Faso (2022-05-15).

### Study design

The present study aimed at measuring plasma biomarkers of *S. haematobium* infection in samples collected during a cross-sectional survey conducted among endemic populations from Burkina Faso, and at performing a comparative evaluation of their diagnostic performance.

The cross-sectional survey was conducted in August 2007 in two rural villages (Barkoumbilen and Barkoundouba) of shrubby savannah areas of Burkina Faso northeast of the capital town Ouagadougou, for malaria epidemiology purposes [34]. A 2 mL venous blood sample was collected from each participant in ethylenediaminetetraacetic acid (EDTA) tubes. Plasma was separated by centrifugation (3min at 2000 rpm), aliquoted and stored at -80˚C until assays were performed. The study area is endemic for *S. haematobium* [1,33].

Assays have been performed on 406 archived plasma specimens to measure the following biomarkers: *Schistosoma* spp. Circulating Anodic Antigen (CAA), *S. haematobium* cell-free circulating DNA (cfDNA), *S. mansoni* cfDNA, IgM and IgG against *S. haematobium* Soluble Worm Adult Protein (SWAP) and Soluble Egg Antigen (SEA). The species-specific results of cfDNA testing were used to confirm that the study area is endemic for *S. haematobium* only. Sample size was determined by the availability of archived plasma specimens and not by statistical power analysis.

The evaluation of the diagnostic performance of the different biomarker tests was conducted comparing the results to those of three different references (i.e. defining true infection status): i) Circulating Anodic Antigen; ii) Composite Reference Standard; iii) Latent Class Analysis.

### Circulating anodic antigen

The Up-Converting Particle Lateral Flow (UCP-LF) assay was conducted in Leiden University Medical Center, the Netherlands, to detect Circulating Anodic Antigen (CAA) in plasma samples as previously described (SCAA20) [16,35]. High Salt Lateral Flow (HSLF) buffer (100 mM HEPES pH 7·5, 270 mM NaCl, 0·5% v/v Tween-20, 1% w/v BSA, 50 μL) was pipetted into the wells of a 96-wells microplate. Each plasma sample (50 μL) was mixed with an equal volume of 4% trichloroacetic acid (TCA) and centrifuged for 5 minutes at 18,000 RCF to separate the supernatant from the pellet. The supernatant (20 μL) and a UCP solution (50 μL) containing

reporter particles (400 nm $Y_2O_2S:Yb^{3+},Er^3$) conjugated with mouse monoclonal anti-CAA antibody (25 μg/mg) were pipetted in the microplate wells. After an incubation of 1 hour at 37˚C with shaking (900 rpm), LF strips with a test line comprising 200 ng mouse monoclonal anti-CAA antibody were added into the wells. A dilution series of a sample with known CAA concentration (pg/ml) was included in the microplate and used as a standard curve to quantify CAA concentrations and to validate the threshold of the assay (10 pg/ml). Strips were incubated overnight and scanned with an UPCON reader (980 nm excitation, 540 nm emission; Labrox Oy, Turku, Finland) for analysis. A threshold of 10 pg/ml (SCAA20, wet test format) was used to define CAA positivity [16,35]. CAA levels were defined based on concentration ranges as follows: level 0 = 0–9 pg/ml range; level 1 = 10–99 pg/ml range; level 2 = 100–999 pg/ml range; level 3 = 1000–9999 pg/ml range; level 4≥10000pg/ml.

## Cell-free circulating DNA

Cell free circulating DNA (cfDNA) was extracted from 100 μL plasma samples using the QIA amp Min Elute cfDNA Mini Kit according to the manufacturer's protocol. Real Time PCR was performed at the Bernard Nocht Institute for Tropical Medicine, Germany, to amplify and detect *S. heamatobium* (*Dra1*) and *S. mansoni* (*Sm1-7*) target sequences (Sh-Forward Primer: 5'-GATCTCACCTATCAGACG AAAC-3', Sh-Reverse Primer: 5'-TCACAACGATACGAC CAAC-3', Sh-Probe: 5'-Joe-TGTTGGAAGZGCCTGTTTCGCAA-BHQ1-3'; Sm-Forward Primer: 5'-CAACCGTTCTATGAA AATCGTTGT-3', Sm-Reverse Primer: 5'-CCA CGCTC TCGCAAATAATCT-3', Sm-Probe: 5'-Fam-TCCGAAACCACTGGACGGATTTTTATG AT-BHQ1-3') as previously described [21, 22]. Primers and probe were included for the detection of Phocid herpesvirus (PhHV) DNA as internal PCR control (PhHV—Forward Primer: 5′ GGG CGA ATC ACA GAT TGA ATC 3′, PhHV—Reverse Primer: 5′ GCG GTT CCA AAC GTA CCA A 3′, PhHV—Probe: 5′ Cy5.5-TTT TTA TGT GTC CGC CAC CA-BBQ 3′). Briefly, the reaction was performed in a total volume reaction of 25 μL containing 12.5 μl HotStarTaq Mastermix (Qiagen, Hilden, Germany), 5 mM MgCl2, 500 nM of each *Schistosoma* primer, 250 nM of each *Schistosoma* probe, 40 nM of the PhHV primer, 50 nM of the PhHV probe, 1.425 μg of the PhHV DNA template, 0.04 μg/μl bovine serum albumin and 5 μL of DNA eluate. The amplification reaction (15 minutes at 95˚C followed by 50 cycles of 15 seconds at 95˚C and 60 seconds at 60˚C with an initial touchdown from 65˚C to 60˚C in the first 11 cycles) was conducted using the Corbett Rotor-Gene 6000 (Qiagen) and results were analysed with the RotorGene 6000 Software v.7.87 (Qiagen). Samples with a clean sigmoid curve within the PCR cycles were considered positive.

## Specific immunoglobulins

Specific IgM and IgG against *S. haematobium* Soluble Worm Adult Protein (SWAP) and Soluble Egg Antigen (SEA) were measured by ELISA using an in house protocol adapted from Mutapi and colleagues [27] as previously described [33]. Antigens were purchased from the Theodor Bilharz Institute, Giza, Egypt. A volume of 100 μL antigen (5μg/mL for SWAP and 10μg/mL for SEA) in coating buffer (0.5 M sodium carbonate pH 9.6) was added per well in 96-wells plates (Nunc Immunosorp), and plates were incubated overnight at 4˚C. After 3 washings (PBS-0.03% Tween 20, PBS-T), 200μL blocking buffer (5% skimmed milk powder in PBS/T) was added per well and plates incubated 2h at room temperature (RT). After 3 washings, plasma samples were added in duplicate in blocking buffer with a total volume of 100μL (1:200 dilution for IgM and 1:100 dilution for IgG) and plates were incubated 2h at RT. After three washings 100 μL HRP-anti-human-IgG (Dako) at 1:1000 dilution in PBS-T was added and plates incubated for 2h at RT. After six washings 100 μL OPD (Sigma) substrate solution

was added, and plates incubated for 20 min in the dark at RT. A volume of 25 µL of $H_2SO_4$ 1 M stop solution was added and plates were read with Synergy HT Biotek reader at the absorbance of 492 nm. A standard curve consisting of a 5-point dilution series of a pool of positive control (patients with positive result of urine filtration microscopy) samples was included in each plate as well as a negative control consisting of a pool of negative control (naïve blood donors) samples and a blank control containing no plasma. The OD value of the blank control was subtracted from the OD values of test samples. The OD values of test samples were then normalised to minimise experimental variation across ELISA plates, by fitting standard curves using least squares minimisation, a three parameters sigmoid model and the Solver plugin in Microsoft Excel (fitted OD, fOD), as previously described [36]. The cutoff used for positivity was the mean absorbance plus 3 standard deviation of negative control samples (cutoff = mean OD neg + 3SD neg).

## Statistical analysis

For binary (i.e. positive/negative) test results, proportions were computed with their 95% Confidence Interval (CI) and shown using barplots. Proportions were compared between population groups by univariate logistic regression, reporting Odds Ratio (OR) with its 95% CI and p-value.

Quantitative (i.e. antibody levels and antigen concentration) test results were normalised by logarithmic transformation. Distributions were compared between population groups by univariate linear regression, reporting the exponential of the beta coefficient with its 95% CI and p-value, and shown using boxplots. Correlation between quantitative test results was displayed using scatter plots and assessed by Pearson test, reporting the rho coefficient and p-value.

The diagnostic performance of each test has been evaluated by comparison of binary results with three references: i) Circulating Anodic Antigen; ii) Composite Reference Standard; iii) Latent Class Analysis. For each comparison, sensitivity and specificity with their respective 95% CI were computed, as well as overall agreement and Cohen's k statistics. Cohen's k was interpreted as follows: values $\leq 0$ as no agreement, 0.01–0.20 as none to slight, 0.21–0.40 as fair, 0.41–0.60 as moderate, 0.61–0.80 as substantial, and 0.81–1.00 as almost perfect agreement.

CAA was chosen as a reference since it is a validated marker of infection [15, 16].

A Composite Reference Standard (CRS) was defined as follows: a sample was considered positive if one or both direct (i.e. CAA and/or cfDNA) test results were positive, while it was considered negative if both direct tests results were negative. The rationale of the algorithm is to define infection status based on direct markers tests, and it assumes that both CAA and cfDNA have 100% specificity [37].

Latent Class Analysis (LCA) is a statistical procedure that is used to identify hidden groups of individuals in a population based on sharing of characteristics. LCA modelling with two clusters was used to define a binary (positive/negative) LCA reference [26]. The model included all available manifest variables: CAA (positive/negative), cfDNA (positive/negative), anti-SWAP IgM level (log10-fOD), anti-SWAP IgG level (log10-fOD), anti-SEA IgG level (log10-fOD), anti-SEA IgM level (log10-fOD). The LCA reference therefore defines *S.haematobium* infection status based on the results of all direct and indirect biomarker tests.

Receiver Operating Characteristic (ROC) analysis was used to evaluate the ability of antibody levels (logfOD) to discriminate between positive and negative reference results, by calculating Area Under Curve values and by computing specificity, sensitivity and overall agreement for each possible cutoff value. An optimal cutoff value was identified by filtering values associated with >75% agreement, sensitivity and specificity, ordering them from higher

to lower agreement, sensitivity and specificity, and selecting the top value. A cutoff Antibody Index (AI) was calculated as the ratio between the value of the cutoff and the value of the negative control.

All analyses were performed in STATA v13.00, except for the LCA analysis that was conducted using R v. 4.3.1 and its package *flexmix*.

## Results

Real Time PCR did detect *S. haematobium* cfDNA in the study sample, but not *S. mansoni* cfDNA, as expected from the literature [1]. Therefore, it can be assumed that CAA in this area is a product of *S. haematobium* and therefore a biomarker of urogenital schistosomiasis. The proportion of positive results for each plasma biomarker of *S. haematobium* infection is shown in Table 1. CAA and cfDNA tests result in an identical proportion of positive subjects, equal to 29%. Specific antibody tests result in a higher proportion of positive subjects, ranging from 47% to 72%, with anti-SEA Ig showing higher positivity than anti-SWAP Ig.

The breadth of antibody response to *S. haematobium* (*Sh*) antigens (i.e. the total number of specific Ig for which a positive result was observed) and the different patterns of positivity are shown in Table 2. Overall, 15% of subjects were seronegative for anti-*Sh* Ig. Among subjects who were seropositive for only one of the four measured Ig (12%), the majority was positive for anti-SEA IgG. Among subjects who were seropositive for two specific Ig (18%), the two most and equally frequent combinations were anti-SWAP IgM/anti-SWAP IgG and anti-SWAP IgG/anti-SEA IgG. Among subjects who were seropositive for three specific Ig (26%), the majority was positive for all Ig except anti-SWAP IgM. Finally, about 29% of subjects were positive for all four specific Ig. In order to investigate the diagnostic potential of indirect markers compared to direct ones, the positivity of CAA and cfDNA direct markers according to the breadth of the anti-*Sh* Ig response was assessed by logistic regression analysis. It can be observed that both CAA (OR = 2.4, 95% CI = 1.9–3.0, p-value<0.001) and cfDNA (OR = 2.6, 95% CI = 2.1–3.4, p-value<0.001) positivity increase significantly with the number of positive specific Ig (Fig 1).

The formal evaluation of the diagnostic performance of the different biomarkers was conducted by comparing each test results to those of three different references (i.e. "true" infection status): i) Circulating Anodic Antigen (CAA); ii) Composite Reference Standard (CRS); iii) Latent Class Analysis (LCA). For each comparison, sensitivity and specificity were calculated as well as the overall agreement.

In comparison with CAA as the reference test (Table 3), cfDNA showed 73% sensitivity and 89% specificity, with a Cohen's k of 0.64, indicating substantial agreement. Indeed, despite resulting in an identical proportion of positive samples, the two tests did not yield identical

**Table 1. Proportion of positive subjects for plasma biomarkers of *S. haematobium* infection.** The table shows, for each plasma biomarker of *S. haematobium* infection, the number of positive subjects (N+) in the study sample (Ntotal = 406), and the proportion of positive subjects (%) with its 95% Confidence Interval (CI).

| Marker | Positive subjects | | |
|---|---|---|---|
| | N+ | Proportion+ (%) | 95% CI |
| CAA | 118 | 29.1 | 24.8–33.7 |
| cfDNA | 118 | 29.1 | 24.8–33.7 |
| anti-SWAP IgM | 192 | 47.3 | 42.5–52.2 |
| anti-SWAP IgG | 223 | 54.9 | 50.0–59.7 |
| anti-SEA IgM | 292 | 71.9 | 67.3–76.1 |
| anti-SEA IgG | 282 | 69.5 | 64.8–73.8 |

**Table 2. Patterns of antibody response against *S. haematobium* antigens.** The table shows the breadth of the antibody response to *S. haematobium* antigens (number of positive anti-*Sh* Ig) and the different positivity patterns of IgM and IgG responses to SWAP and SEA antigens, with their relative proportion (%) in the total study population. Antibody positivity is indicated by a coloured cell (anti-SWAP IgM: pink; anti-SWAP IgG: yellow; anti-SEA IgM: violet; anti-SEA IgG: orange).

| Breadth | | Pattern | | | | |
|---|---|---|---|---|---|---|
| anti-*Sh* Ig+ (N) | Proportion (%) | anti-SWAP | | anti-SEA | | Proportion (%) |
| | | IgM | IgG | IgM | IgG | |
| 0 | 14.5 | | | | | 14.5 |
| 1 | 12.3 | ■(pink) | | | | 2.2 |
| | | | ■(yellow) | | | 0.3 |
| | | | | ■(violet) | | 4.7 |
| | | | | | ■(orange) | 5.2 |
| 2 | 17.5 | ■(pink) | ■(yellow) | | | 5.9 |
| | | ■(pink) | | ■(violet) | | 1.2 |
| | | ■(pink) | | | ■(orange) | 1.5 |
| | | | ■(yellow) | ■(violet) | | 3.7 |
| | | | ■(yellow) | | ■(orange) | 5.2 |
| | | | | ■(violet) | ■(orange) | 1.5 |
| 3 | 26.4 | ■(pink) | ■(yellow) | ■(violet) | | 1.0 |
| | | ■(pink) | ■(yellow) | | ■(orange) | 6.2 |
| | | | ■(yellow) | ■(violet) | ■(orange) | 17.7 |
| 4 | 29.3 | ■(pink) | ■(yellow) | ■(violet) | ■(orange) | 29.3 |
| Total | 100 | Total | | | | 100 |

results (S1 Fig). The comparison of CAA and cfDNA results stratified by CAA level shows that the proportion of "false negative" cfDNA results decreases significantly with increasing CAA levels (OR = 0.23, 95% CI = 0.17–0.31, p-value<0.001; Fig 2). Among specific antibodies, Cohen's k ranged from 0.09 to 0.37, indicating none to fair agreement, with the best performance being observed for anti-SWAP IgG, showing 88% sensitivity and 59% specificity.

A CRS was built using an algorithm that classified a sample as positive if it exhibits positive results of either or both direct marker tests (CAA+ OR cfDNA+), and as negative if they exhibit negative results of both direct marker tests (CAA- AND cfDNA-), irrespectively of the

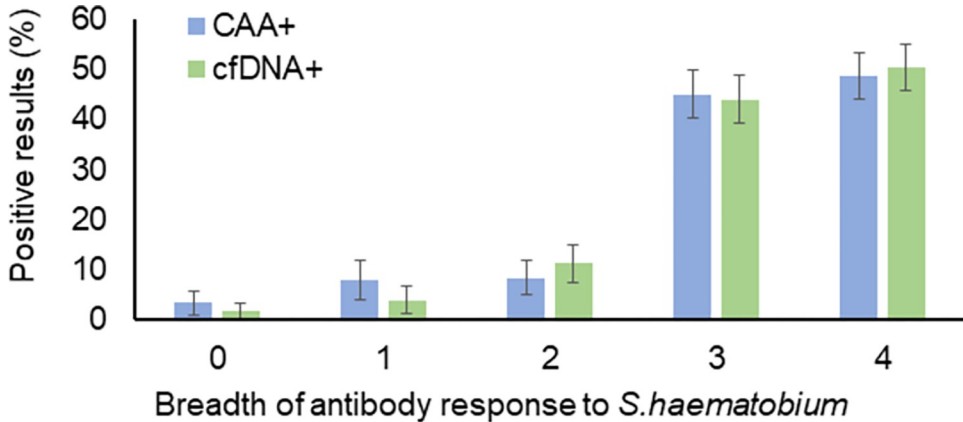

**Fig 1. CAA and cfDNA positivity according to the breadth of the antibody response to *S. haematobium* antigens.** The figure shows the proportion of subjects showing results to CAA (blue) and cfDNA (green) testing according to the breadth of the antibody response to *S. haematobium* antigens. Bars represent proportion and whiskers represent the standard error of the proportion.

**Table 3. Diagnostic performance of plasma biomarkers compared to Circulating Anodic Antigen.** The table shows, for each plasma biomarker of *S. haematobium* infection, the sensitivity (Se, %) and specificity (Sp, %) with their 95% Confidence Interval (CI), the overall agreement (%) and the Cohen's K value, when compared to Circulating Anodic Antigen as the reference.

| Marker | Circulating Anodic Antigen | | | |
|---|---|---|---|---|
| | Se (95% CI) | Sp (95% CI) | Agreement | Cohen's K |
| CAA | - | - | - | - |
| cfDNA | 73 (65–81) | 89 (85–93) | 84 | 0.62 |
| anti-SWAP IgM | 55 (46–64) | 56 (50–62) | 56 | 0.09 |
| anti-SWAP IgG | 88 (82–94) | 59 (53–64) | 67 | 0.37 |
| anti-SEA IgM | 94 (90–98) | 37 (32–43) | 54 | 0.22 |
| anti-SEA IgG | 95 (91–99) | 41 (35–47) | 57 | 0.25 |

results of indirect marker tests. This CRS resulted in a 37% proportion of positive subjects. In comparison with the CRS (Table 4), both CAA and cfDNA showed 79% sensitivity and 100% specificity (as expected), with an overall almost perfect agreement (92%, Cohen's K = 0.82). Among specific antibodies, Cohen's k ranged from 0.11 to 0.43, indicating slight to moderate agreement, and once again the best performance was observed for anti-SWAP IgG, showing 85% sensitivity and 63% specificity.

Finally, result of each biomarker were compared with results of LCA modelling (Table 5), where a sample was classified as positive or negative based on all available biomarker data, and that resulted in a 35% proportion of positive subjects. In comparison with the LCA, both CAA and cfDNA showed substantial agreement, with Cohen's k equal to 0.62 and 0.75 respectively. Among specific antibodies, Cohen's k ranged from 0.15 to 0.55, indicating slight to moderate agreement. Consistently with previous comparisons, the best performance was observed for anti-SWAP IgG, showing 97% sensitivity and 67% specificity.

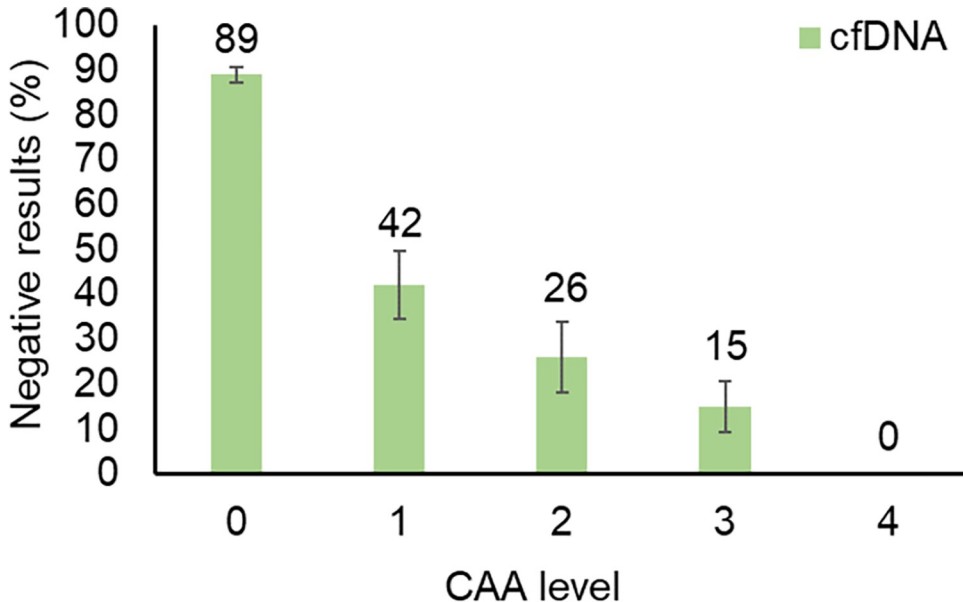

**Fig 2. Proportion of cfDNA negative results according to CAA levels.** The figure shows the proportion of negative results according to CAA levels, i.e. concentration ranges (level 0 = 0–9 pg/ml range; level 1 = 10–99 pg/ml range; level 2 = 100–999 pg/ml range; level 3 = 1000–9999 pg/ml range; level 4≥10000pg/ml). Bars represent proportion and whiskers represent the standard error of the proportion.

**Table 4. Diagnostic performance of plasma biomarkers compared to a Composite Reference Standard.** The table shows, for each plasma biomarker of *S. haematobium* infection, the sensitivity (Se, %) and specificity (Sp, %) with their 95% Confidence Interval (CI), the overall agreement (%) and the Cohen's K value, when compared to a Composite Reference Standard.

| Marker | Composite Reference Standard | | | |
|---|---|---|---|---|
| | Se (95% CI) | Sp (95% CI) | Agreement | Cohen's K |
| CAA | 79 (72–85) | 100 (100–100) | 92 | 0.82 |
| cfDNA | 79 (72–85) | 100 (100–100) | 92 | 0.82 |
| anti-SWAP IgM | 55 (47–63) | 57 (51–63) | 56 | 0.11 |
| anti-SWAP IgG | 85 (79–90) | 63 (57–68) | 71 | 0.43 |
| anti-SEA IgM | 94 (90–98) | 41 (35–47) | 61 | 0.29 |
| anti-SEA IgG | 94 (90–98) | 45 (39–51) | 63 | 0.33 |

Receiver Operating Curve (ROC) analysis was conducted to further evaluate the diagnostic performance of *S. haematobium* specific antibodies. The Area Under Curve (AUC) statistics, a measure of the ability of a marker to correctly classify positive and negative cases as defined by a reference standard, was computed for each antibody test against the three references (CAA, CRS, LCA). The comparison of AUC allows to identify the marker showing the best diagnostic performance: the highest AUC value was observed for anti-SEA IgG, followed closely by anti-SWAP IgG, for each reference (Table 6 and S2–S4 Figs).

For anti-SWAP and anti-SEA IgG, ROC analysis was employed for further comparison with CAA to generate Sensitivity/Specificity plots (Fig 3), which show that selecting a higher cutoff value compared to the analytical one would result in better diagnostic performances. For instance, for anti-SWAP IgG, a cutoff of 0.196 log(fOD) corresponding to a 2.3 Antibody Index (AI) would result in 75% sensitivity, 76% specificity and 76% agreement. Similarly, for anti-SEA IgG, a cutoff of 0.242 log(fOD) corresponding to a 4.5 AI would result in 82% sensitivity, 80% specificity and 81% agreement. Also, the distribution of antibody levels according to CAA positivity was investigated, as well as the correlation between antibody levels and CAA concentration ranges. The distribution of antibody levels was significantly higher in subjects that were positive to CAA than in negative subjects, for both anti-SWAP IgG (Exp$\beta$ = 15.3, 95% CI = 9.8–23.8, p-value<0.001; Fig 4, left panel) and anti-SEA IgG (Exp$\beta$ = 13.3, 95% CI = 9.4–18.8, p-value<0.001; Fig 4, right panel). Finally, a scarce albeit significant positive correlation was observed between antibody levels and CAA concentration, both for anti-SWAP IgG (Pearson P-value<0.001, rho = 0.407; Fig 5 left panel) and anti-SEA IgG (Pearson P-value<0.001, rho = 0.396 Fig 5, right panel).

**Table 5. Diagnostic performance of plasma biomarkers compared to Latent Class Analysis classification.** The table shows, for each plasma biomarker of *S. haematobium* infection, the sensitivity (Se, %) and specificity (Sp, %) with their 95% Confidence Interval (CI), the overall agreement (%) and the Cohen's K value, when compared to Latent Class Analysis classification.

| Marker | Latent Class Analysis | | | |
|---|---|---|---|---|
| | Se (95% CI) | Sp (95% CI) | Agreement | Cohen's K |
| CAA | 67 (60–75) | 92 (89–95) | 83 | 0.62 |
| cfDNA | 76 (69–83) | 97 (94–99) | 89 | 0.75 |
| anti-SWAP IgM | 58 (50–66) | 58 (52–64) | 58 | 0.15 |
| anti-SWAP IgG | 95 (92–99) | 67 (61–73) | 77 | 0.55 |
| anti-SEA IgM | 99 (97–100) | 43 (37–49) | 63 | 0.34 |
| anti-SEA IgG | 100 (100–100) | 47 (41–53) | 66 | 0.39 |

**Table 6. Diagnostic performance of antibody tests resulting from ROC analysis.** The table shows, for each *S. haematobium* antibody test, the Area Under Curve (AUC) resulting from ROC analysis conducted to evaluate the diagnostic performance compared to Circulating Anodic Antigen (CAA), Composite Reference Standard (CRS), and Latent Class Analysis (LCA) classifications.

| Marker | Area Under Curve (AUC) | | |
|---|---|---|---|
| | CAA | CRS | LCA |
| anti-SWAP IgM | 0.567 | 0.597 | 0.635 |
| anti-SWAP IgG | 0.826 | 0.832 | 0.915 |
| anti-SEA IgM | 0.765 | 0.796 | 0.861 |
| anti-SEA IgG | 0.878 | 0.901 | 0.995 |

## Discussion

In summary, the measurement of plasma biomarkers of *S. haematobium* infection among 406 subjects living in rural endemic villages of Burkina Faso, West Africa, has shown that both CAA and cfDNA testing result in a 29% proportion of positive samples. A 29% prevalence of urogenital schistosomiasis in this area is in line with an estimate for Burkina Faso based on systematic review of survey data and geostatistical modelling (23%) [1]. However, it likely represents an underestimation, as suggested by the observation that CRS and LCA show a proportion of positive subjects of 37% and 35%, respectively. Indeed, despite an identical proportion of positive samples identified by CAA and cfDNA testing, the two tests did not yield identical results. Discordant CAA negative/cfDNA positive results could be attributed to differences in the stage specificity of the two markers, with cfDNA detecting all stages including larval ones while CAA detecting adults [38], and/or by recently cleared infections with cfDNA persisting in circulation longer than CAA [19]. On the other hand, discordant CAA positive/cfDNA negative results could be attributed to a suboptimal cfDNA sensitivity at low CAA concentrations because of an insufficient plasma volume used for nucleic acid extraction in this study (1000μl of plasma was used in a previous work [22] compared to the 100μl used in the present study), an hypothesis supported by the observation that the proportion of "false negative" cfDNA results decreases significantly with increasing CAA levels. Despite some discordant results, there was a substantial agreement between CAA and cfDNA testing, along with a comparable agreement with the results of CRS and LCA (Fig 6). Taken together, the identical proportion of positive samples identified by the two tests, and the substantial agreement of the

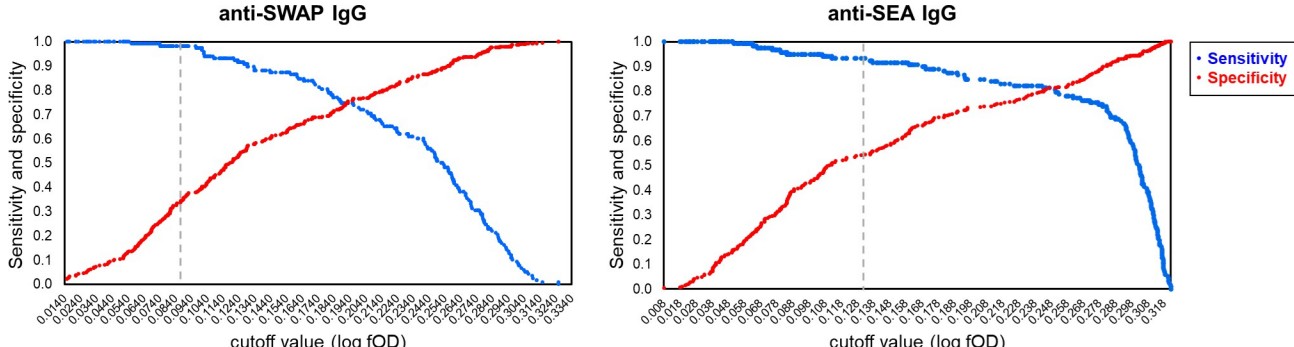

**Fig 3. Cutoff levels of anti-SWAP and anti-SEA IgG and diagnostic performance compared to Circulating Anodic Antigen.** The figure shows the plot of sensitivity (blue) and specificity (red) compared to Circulating Anodic Antigen for each cutoff value of anti-SWAP IgG (left panel) and anti-SEA IgG (right panel). The grey vertical line indicates the analytical cutoff.

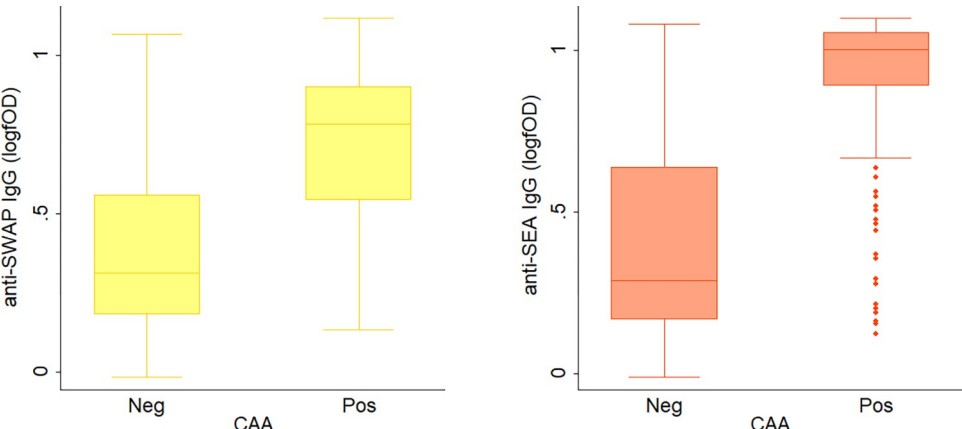

**Fig 4. Distribution of anti-SWAP and anti-SEA IgG levels according to Circulating Anodic Antigen positivity.**
The figure shows the boxplot distribution of anti-SWAP (left panel) and anti-SEA (right panel) IgG levels according to
Circulating Anodic Antigen (CAA) positivity (negative subjects, Neg; positive subjects; Pos). The horizontal line
represents the 50% percentile (median), the box lower and upper limits represent the 25% and 75% percentiles
respectively, and lower and upper whiskers represent the 5% and 95% percentiles respectively, while the dots are
outliers of the distribution.

results, indicate that CAA and cfDNA are equivalent good plasma biomarkers of infection in
areas of *S. haematobium* endemicity.

As expected, specific antibody testing resulted in a much higher proportion of positive sam-
ples (47%-72% range), as the presence of specific antibodies might reflect not only current
infection but past exposure to the parasite. Interestingly, it was observed that the breadth of
the antibody response to *S. haematobium* was positively associated with both CAA and cfDNA
positivity, indicating that subjects harbouring antibodies against multiple *S. haematobium*
antigens are more likely to be currently infected. The analysis of individual responses has
shown that IgG against SWAP and SEA exhibit a higher agreement with the three references
(CAA, CRS and LCA) compared to IgM. When compared to CAA, ROC analysis confirmed a
better ability of IgG to distinguish positive and negative subjects, and showed that higher cutoff

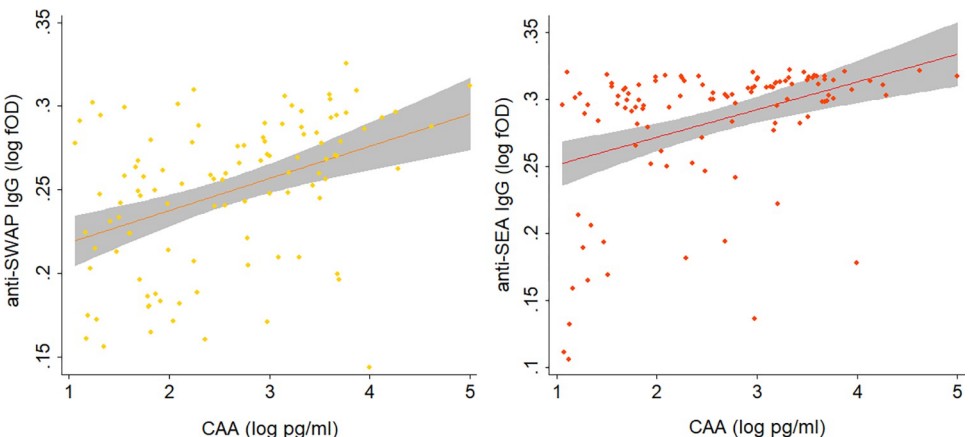

**Fig 5. Correlation of anti-SWAP and anti-SEA IgG levels with Circulating Anodic Antigen concentration.** The
figure shows scatterplots illustrating the correlation of anti-SWAP IgG levels (log fOD; left panel) and anti-SEA IgG
levels (log fOD; right panel) with CAA concentration (log pm/ml). Dots represent actual data points, the line represent
the fitted data, and the grey area represent the 95% confidence interval of the fitted line.

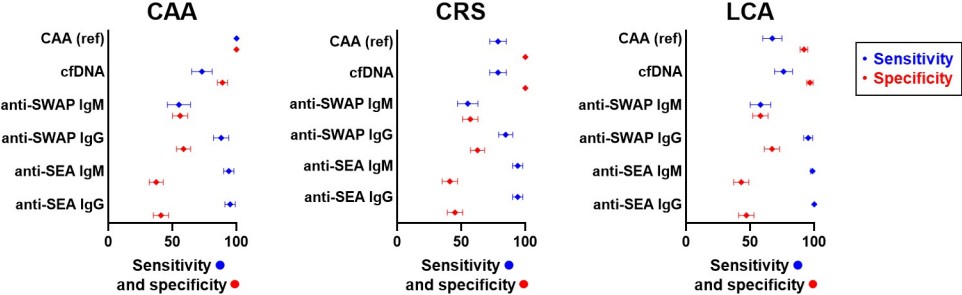

**Fig 6. Sensitivity and specificity of *S. haematobium* plasma biomarkers compared to Circulating Anodic Antigen, Composite Reference Standard and Latent Class Analysis.** The figure shows forest plots of sensitivity (blue) and specificity (red) of different *S. haematobium* plasma biomarkers compared to Circulating Anodic Antigen (CAA, left panel), Composite Reference Standard (CRS, mid panel) and Latent Class Analysis (LCA, right panel). Dots represent the estimate and whiskers represent the 95% CI of the estimate.

values result in greatly enhanced specificity and overall diagnostic performance of these indirect biomarkers. Indeed, IgG levels were significantly higher in CAA positive than CAA negative subjects, indicating that antibody levels are increased in subjects with current infection compared to subjects with past exposure, and highlighting the relevance of the choice of appropriate cutoff values for accurate detection of infection. However, only a moderate correlation between IgG antibody levels and CAA concentration was observed, indicating that IgG levels cannot be used as markers of *S. haematobium* infection intensity.

One limitation of the present study is the lack of results from urine filtration microscopy, as urine specimens were not collected during the original investigation survey. Also, the antibody response to CAP was not investigated, as this antigen could not be procured at the time of analysis. Since CAP is a cercarial antigen, data would have been beneficial for interpreting discordant results between cfDNA and CAA testing, by either supporting or refusing the hypothesis that very recent infections with larval stages would be detected by cfDNA but not CAA.

Despite the above limitations, the comparative evaluation of plasma biomarkers presented herein represents, to our knowledge, the first conducted among endemic populations. The data provide valuable insights that can inform the choice of testing to be employed in areas endemic for urogenital schistosomiasis, depending on the purpose, context, needs and available laboratory resources. For diagnostic "test-to-treat" purposes, when infection with *S. haematobium* is suspected, either CAA or cfDNA testing can be performed for confirmation, even in the absence of urine filtration microscopy, while anti-SWAP and anti-SEA IgG testing could be performed for supporting evidence. A similar strategy is currently recommended in non-endemic countries for the diagnosis of schistosomiasis [39]. It is noteworthy that only cfDNA is a species-specific marker, and therefore CAA positive results do not exclude an infection with other *Schistosoma* species. However, only CAA provides a quantitative measurement (i.e. CAA concentration) associated with worm burden that can be used for follow-up of treated patients [40]. Likewise, for epidemiological investigations, either CAA or cfDNA testing can be performed to assess infection in areas of *S. haematobium* endemicity depending on laboratory resources, since they are equivalent good markers. Unfortunately, neither of the two methods is currently suitable for point-of-care diagnosis in resource limited settings. As mentioned above, it should also be considered that only cfDNA testing would provide a confirmation of the species while only CAA testing results can be analyzed as a quantitative variable which might be an advantage for association studies. Finally, for surveillance purposes, once the appropriate cutoff has been established for a given epidemiological context to ensure optimal agreement with markers of current infection, anti-SWAP or anti-SEA IgG testing could

be either employed, with the important advantages of allowing multiplexing with other antibody tests as well as automation.

The availability of different accurate diagnostic tests for *S. haematobium* infection based on the detection of plasma biomarkers represents a significant development in the control of urogenital schistosomiasis in endemic areas.

## Supporting information

**S1 Fig. Venn proportional diagram showing the comparison of CAA and cfDNA testing results.**
(PDF)

**S2 Fig. ROC curves illustrating the diagnostic performance of anti-*S.haematobium* antibodies compared to Circulating Anodic Antigen.**
(PDF)

**S3 Fig. ROC curves illustrating the diagnostic performance of anti-*S.haematobium* antibodies compared to Composite Reference Standard.**
(PDF)

**S4 Fig. ROC curves illustrating the diagnostic performance of anti-*S.haematobium* antibodies compared to Latent Class Analysis.**
(PDF)

## Acknowledgments

The authors would like to thank participants for making the study possible; physicians, nurses, laboratory technicians, data management staff at Centre National de Recherche et Formation sur le Paludisme for their skilled work; Zeno Bisoffi at Centre for Tropical Diseases in Negrar, Italy, for contributing sera samples of patients infected with *S. haematobium*; Carlota Dobano, Gemma Moncunill and colleagues at Barcelona Institute for Global Health, Spain, for critical discussion of the results; the Italian Network for Neglected Tropical Diseases for promoting research, control and advocacy activities for the fight against NTDs.

## Author Contributions

**Conceptualization:** Daniela Fusco, David Modiano, Fabrizio Bruschi, Valentina D. Mangano.

**Data curation:** Valentina D. Mangano.

**Formal analysis:** Marco Fornili, Laura Baglietto, Valentina D. Mangano.

**Investigation:** Mireille Ouedraogo, Jana Christina Hey, Stan Hilt, Veronica Rodriguez Fernandez, Doris Winter, Ravo Razafindrakoto, Youssouf Kabore.

**Methodology:** Valentina D. Mangano.

**Resources:** Issa Nebie, Govert J. van Dam, Paul L. A. M. Corstjens, Daniela Fusco, David Modiano, Fabrizio Bruschi.

**Supervision:** Valentina D. Mangano.

**Writing – original draft:** Valentina D. Mangano.

**Writing – review & editing:** Mireille Ouedraogo, Jana Christina Hey, Stan Hilt, Veronica Rodriguez Fernandez, Pytsje T. Hoekstra, Govert J. van Dam, Paul L. A. M. Corstjens, Daniela Fusco, David Modiano, Fabrizio Bruschi, Valentina D. Mangano.

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
