## [Decision Letter · Decision Letter 0]

22 Jul 2024

Dear Dr. Mangano,

Thank you very much for submitting your manuscript "Comparative evaluation of plasma biomarkers of Schistosoma haematobium infection in endemic populations from Burkina Faso" for consideration at PLOS Neglected Tropical Diseases. As with all papers reviewed by the journal, your manuscript was reviewed by members of the editorial board and by several independent reviewers. The reviewers appreciated the attention to an important topic. Based on the reviews, we are likely to accept this manuscript for publication, providing that you modify the manuscript according to the review recommendations. 

Sincerely,

Uwem Friday Ekpo, PhD

Academic Editor

Eva Clark

Section Editor

Reviewer's Responses to Questions

**Key Review Criteria Required for Acceptance?**

**Methods**

-Are the objectives of the study clearly articulated with a clear testable hypothesis stated?

-Is the study design appropriate to address the stated objectives?

-Is the population clearly described and appropriate for the hypothesis being tested?

-Is the sample size sufficient to ensure adequate power to address the hypothesis being tested?

-Were correct statistical analysis used to support conclusions?

-Are there concerns about ethical or regulatory requirements being met?

Reviewer #1: -Are the objectives of the study clearly articulated with a clear testable hypothesis stated?

Yes

-Is the study design appropriate to address the stated objectives?

Yes

-Is the population clearly described and appropriate for the hypothesis being tested?

Yes

-Is the sample size sufficient to ensure adequate power to address the hypothesis being tested?

Yes

-Were correct statistical analysis used to support conclusions?

Yes

-Are there concerns about ethical or regulatory requirements being met?

No

Reviewer #2: The objectives of the study are clearly stated. The study design is appropriate. The population is clearly described and appropriate. Correct statistical methods were used. There are no ethical or regulatory requirement concerns.

A few suggestions …

The authors should consider providing a power analysis or mention their sample size calculations.

Line 257: Please specify the type of confidence intervals used.

Line 258: Did you fit unadjusted and adjusted logistic regression models? Please specify the variables considered and/or used in the final model(s). Did you obtain/report odds ratios? Please clarify.

Lines 259-262: Why did you use Spearman (for ranked data) as opposed to Pearson (for normal data) to examine correlations with antibody levels and antigen concentration? You then used linear regression to examine antibody levels and antigen concentration, which requires normal data. If these variables were transformed to normalize them, please specify so here. What variables were considered for the linear regression models?

**Results**

-Does the analysis presented match the analysis plan?

-Are the results clearly and completely presented?

-Are the figures (Tables, Images) of sufficient quality for clarity?

Reviewer #1: -Does the analysis presented match the analysis plan?

Yes

-Are the results clearly and completely presented?

Yes

-Are the figures (Tables, Images) of sufficient quality for clarity?

Yes

Reviewer #2: The results match the analysis plan for the most part. The results are clearly presented. Most tables and figures are appropriate. A few suggestions …

Table 1: Consider including the total N in a table footnote so you can delete the N total column (feels unnecessary to have a whole column for a single number).

Lines 322-325: Are these unadjusted or adjusted logistic regressions models? Are there any variables you could consider in adjusted models? If so, a table of the results would be needed. Throughout the paper, P-values not really needed since you report 95% CI, which show the significance. 

For all tables with 95% CIs, consider combining the statistic with the corresponding CI so you do not have multiple LCL and UCL column labels. For example, in Table 3, you could use Se (95% CI), Sp (95% CI), Agreement, Cohen’s K as the columns.

Did you actually use any linear regression models? No related results were reported in the text.

**Conclusions**

-Are the conclusions supported by the data presented?

-Are the limitations of analysis clearly described?

-Do the authors discuss how these data can be helpful to advance our understanding of the topic under study?

-Is public health relevance addressed?

Reviewer #1: -Are the conclusions supported by the data presented?

Yes

-Are the limitations of analysis clearly described?

Yes

-Do the authors discuss how these data can be helpful to advance our understanding of the topic under study?

Yes

-Is public health relevance addressed?

Yes

Reviewer #2: Conclusions are supported by the data presented. Limitations are clear. Authors do discuss public health relevance.

**Editorial and Data Presentation Modifications?**

Reviewer #1: See attached document

Reviewer #2: Please review once more for typos such as the following …

Line 87: ‘generation’ should be plural

Line 167: remove ‘)’ before the period

**Summary and General Comments**

Reviewer #1: Strength

One of the challenges in the develop of high sensitive and specific point of care diagnostic tools which can be used low resource endemic areas is the lack of appropriate diagnostic tools. The Composite Reference Standard

(CRS) and Latent Class Analysis (LCA) methods described in the manuscript by Mireille Ouedraogo and colleagues can be used as references for new diagnostic tools for NTDs. 

Limitation

Considering that schistosomiasis is mostly endemic in poor resource areas the techniques described in this manuscript will not be applicable in these settings. The techniques described are expensive, they require sophisticated laboratories and trained personnel among other things especially the PCR methods so the is still need to develop tools for use in poor endemic settings.

Reviewer #2: Overall the paper is well written, and the public health relevance is clear. The statistical methods and overall conclusions are appropriate. Addressing some of the gaps mentioned in the methods and results will strengthen the manuscript even more.

PLOS authors have the option to publish the peer review history of their article (what does this mean?). If published, this will include your full peer review and any attached files.

Reviewer #1: No

Reviewer #2: No

Figure Files:

Data Requirements:

Reproducibility:

References

---

## [Editor Report · Decision Letter 1]

5 Aug 2024

Dear Dr. Mangano,

We are pleased to inform you that your manuscript 'Comparative evaluation of plasma biomarkers of Schistosoma haematobium infection in endemic populations from Burkina Faso' has been provisionally accepted for publication in PLOS Neglected Tropical Diseases.

Best regards,

Jong-Yil Chai

Section Editor

Jong-Yil Chai

Section Editor

Your revised manuscript is now acceptable by PLoS NTD.

---

## [Editor Report · Acceptance letter]

12 Sep 2024

Dear Dr. Mangano,

We are delighted to inform you that your manuscript, "Comparative evaluation of plasma biomarkers of Schistosoma haematobium infection in endemic populations from Burkina Faso," has been formally accepted for publication in PLOS Neglected Tropical Diseases.

Best regards,

Shaden Kamhawi

co-Editor-in-Chief

Paul Brindley

co-Editor-in-Chief
